# Bridging Gaps in Content and Knowledge for Multimodal Entity Linking

## ABSTRACT

Multimodal Entity Linking (MEL) aims to address the ambiguity in multimodal mentions and associate them with Multimodal Knowledge Graphs (MMKGs). Existing works primarily focus on designing multimodal interaction and fusion mechanisms to enhance the performance of MEL. However, these methods still overlook two crucial gaps within the MEL task. One is the content discrepancy between mentions and entities, manifested as uneven information density. The other is the knowledge gap, indicating insufficient knowledge extraction and reasoning during the linking process. To bridge these gaps, we propose a novel framework FissFuse, as well as a plug-and-play knowledge-aware re-ranking method KAR. Specifically, FissFuse collaborates with the Fission and Fusion branches, establishing dynamic features for each mention-entity pair and adaptively learning multimodal interactions to alleviate content discrepancy. Meanwhile, KAR is endowed with carefully crafted instruction for intricate knowledge reasoning, serving as re-ranking agents empowered by Large Language Models (LLMs). Extensive experiments on two well-constructed MEL datasets demonstrate outstanding performance of FissFuse compared with various baselines. Comprehensive evaluations and ablation experiments validate the effectiveness and generality of KAR.

## CCS CONCEPTS

• **Information systems** → **Multimedia information systems**; **Multimedia databases**; **Information retrieval**.

## KEYWORDS

Multimodal Entity Linking, Multimodal Knowledge Graph, Multimodal Fusion, Content Discrepancy

## 1 INTRODUCTION

**Multimodal Entity Linking (MEL)**, playing a crucial role in associating internet content with multimodal knowledge graphs (MMKGs) [7, 16, 22, 33], has garnered increased attention and facilitated numerous knowledge-intensive applications, such as visual question answering [19, 26] and semantic search [15]. Compared with traditional text-based EL task [4, 8], MEL could effectively leverage visual cues to alleviate the issue of mention ambiguity. For instance, although Figure 1 illustrates that the mention "Hussein" in the sentence could refer to multiple entities, such as "Hussein of

**Unpublished working draft. Not for distribution.**

**Multimodal Mention**

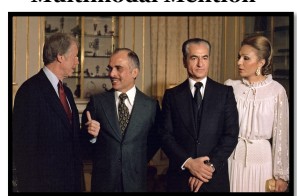
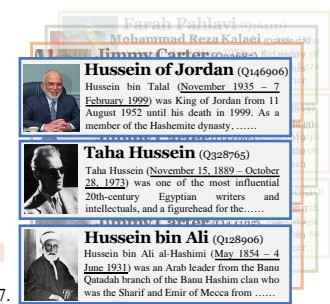

Hussein with American president Jimmy Carter, Iranian Shah Mohammad Reza and Shahbanu Farah, 31 December 1977.

**Figure 1: Illustration of MEL task. Left: multimodal mention. Right: candidate entities.**

Jordan" and "Taha Hussein", with the assistance of visual information, one can more easily discern that the entity "Hussein of Jordan" better aligns with "Hussein".

Therefore, for the MEL task, integrating visual/textual features is an intuitive and effective approach for disambiguation [1, 20]. Towards this goal, a series of research efforts explore how to leverage the interaction and fusion of multimodal features [18, 38, 41] to achieve more precise disambiguation. Despite steady progress and performance improvements demonstrated in various benchmarks [34, 35], however, these studies unintentionally overlook two crucial issues:

- **Content Discrepancy.** As depicted in Figure 1, the brevity and contextual conciseness of mentions sharply contrast with the detailed and exhaustively descriptive nature of entities, for both textual and visual cues. Moreover, the mention images encompass a broader context by including multiple objects and scenes, whereas entities require a more centralized focus on a specific object. In this case, the disparity in content causes severe information mismatch between the scene-aware mention images and object-centric entities, which in turn undermines the effectiveness of fusion-based approaches. Consequently, this poses a significant challenge in dynamically and adaptively adjusting to the imbalanced multimodal information presented in mentions and entities.

- **Insufficient Knowledge Utilization.** Most existing MEL methods usually employ pre-trained language models, such as BERT [5], to encode textual contents before measuring mention-entity alignment. They are confined to encoding lengthy textual content without sufficient knowledge reasoning, resulting in the potential loss of crucial information. For example, if one notices the underlined dates in Figure 1, it is easy to eliminate candidate entities "Taha Hussiein" and "Hussein bin Ali". Although the recent success in Large Language Models (LLMs) [23, 30, 31] showcasing capabilities in knowledge storage and reasoning, these capabilities are not fully utilized. Therefore, it is necessary to design a LLM-oriented knowledge-aware strategy to support the sufficient knowledge reasoning.

To narrow the gaps and address the challenges, in this paper, we propose a novel MEL method FissFuse as well as a plug-and-play knowledge-aware re-ranking strategy KAR. Specifically, FissFuse begins with extracted unified multimodal representations of mentions and entities. Then, in two branches, namely Fission Branch and Fusion Branch, we dynamically and adaptively enrich the representations in different views of interaction. In the Fission branch, we utilize cross-attention mechanism to flexibly establish dynamic features for each mention-entity pair, allowing the same mention to dynamically vary with different entities, and vice versa for entities in various mention contexts. As for the Fusion branch, we adaptively construct fused representations, leveraging semantically rich modality to compensate for deficient one. The two branches jointly encourage ample feature intertwining across different scenarios. Finally, we regard LLMs as re-ranking agents. We devise a universal re-ranking strategy to optimize initial entities ranking by fully leveraging the internal entity knowledge and reasoning capability of LLMs. In summary, our main contributions are threefold:

- We investigate the issue of content discrepancy in the MEL task and propose the FissFuse framework, which can dynamically and adaptively fuse features of different modalities, sources, and granularities for disambiguation.
- To address the issue of knowledge gap, we devise a plug-and-play re-ranking module KAR. It leverages the reasoning capabilities of LLMs as a re-ranking agent, enabling seamless integration with existing LLMs to improve MEL task performance.
- Experimental results show that our approach achieves state-of-the-art performance on prevalent datasets. Comprehensive ablation studies further validate the effectiveness and generality of the proposed FissFuse and KAR.

## 2 RELATED WORK

**Neural Entity Linking.** Neural entity linking [27] is aim to disambiguate mentions and associate them with entities of knowledge base. The early methods focused on text modality and utilized pre-trained embedding to capture relevance between mention and entity with CNN and LSTM [10, 11, 29]. With the popularity of Transformer [32], many research emerged and these methods could be divided into two steams based on the scope of contextual information: local and global. The former methods disambiguate mention based on its surrounding sentence. They utilized pre-trained BERT [5] with knowledge attention [24], proposed two-state encoders [37], and incorporated generative model BART [13] into entity linking [3]. The latter consider relationship among mentions and disambiguate them collectively via converting it into a sequence decision problem [6] and introducing GCN architecture to collectively identify the mappings [36]. However, these methods cannot leverage visual information for disambiguation, limiting their performance in multimodal contexts. The pioneering research from Moon et al. [21] proposed integrating visual features into the entity linking process to mitigate ambiguity in social media. Subsequently, research focused on multimodal fusion *within* the mention and entity context for different goals via concatenation [1, 42], diverse attention mechanisms [34, 40, 41]. These methods unintentionally ignore the interaction *between* mentions and entities. Recent

studies [18, 38] explored dynamic feature interaction between multimodal features. MIMIC [18] designed three interaction units to model the feature interaction between mention-entity and textual-visual, but its focused is on using entity attributes to complement short texts and implicit visual cues. This is significantly different from the content discrepancy we want to explore. DRIN [38] is the most relevant work to our research. It used graph neural networks to model the interactions between different mentions and entities. However, they did not explore how to leverage fine-grained features, which may lead to suboptimal results. Our work enables adaptive interaction among various modalities, sources, and granularities of features, setting it apart from these methods.

**Large Language Models.** Recently, LLMs [31, 43] have become a major research focus, demonstrating powerful capabilities in content comprehension and reasoning. For MEL task, [28] explored visual instruction fine-tuning and constrained decoding for generative disambiguation. However, how to seamlessly integrate existing language models into MEL tasks and fully leverage their advantages, while avoiding expensive and cumbersome fine-tuning, remains to be explored.

## 3 METHODOLOGY

### 3.1 Problem Definition

We formulate the MEL task following the previous works [34, 38]. Formally, given a multimodal mention $M_i$ composed of a sentence $m_t$ and an image $m_v$, there is a set of candidate entities $C(M_i) = \left\{E_j = (e_t, e_v)\right\}_{j=1}^{N}$, where $e_t$ represents entity description, $e_v$ denotes entity image and $N$ is the number of candidate entities. MEL task aims to retrieve the ground truth entities $E_i^*$ based on the similarity between mention and each candidate entity, i.e,

$$E_i^* = \underset{E_j \in C(M_i)}{\arg\max} \ \text{sim}_\theta(M_i, E_j), \quad (1)$$

where $\text{sim}_\theta(\cdot, \cdot)$ is the similarity function and $\theta$ represents the learnable parameter of the function.

### 3.2 Feature Encoding

We start with feature encoding to introduce how we construct $\text{sim}_\theta(\cdot, \cdot)$ function. We first extract textual and visual features for both mentions and entities with pre-trained models. Specifically, we employ a frozen text encoder $\text{Enc}^T(\cdot)$ and a frozen image encoder $\text{Enc}^V(\cdot)$. We add trainable linear layers with layer normalization [2] to convert the dimension of the features. This process can be illustrated as,

$$\begin{aligned} \mathbf{F}_m^T, \mathbf{F}_m^V &= \text{Enc}^T(m_t), \text{Enc}^V(m_v), \\ \mathbf{F}_e^T, \mathbf{F}_e^V &= \text{Enc}^T(e_t), \text{Enc}^V(e_v). \end{aligned} \quad (2)$$

Here, $\mathbf{F}_m^T, \mathbf{F}_e^T \in \mathbb{R}^{L \times d}$ represent sequential text features for the mention sentence and entity description, respectively, and $T$ is text length. Similarly, $\mathbf{F}_m^V, \mathbf{F}_e^V \in \mathbb{R}^{P \times d}$ denote image patch features for mention and entity respectively, and $P$ is the number of image patches. The features can be regarded as fine-grained local features. In addition, it is crucial to obtain coarse-grained global features for a comprehensive understanding of semantics. This can be achieved using special tokens ([CLS] and [EOS]) of the encoders, or applying mean pooling over the hidden states. Consequently, we obtain

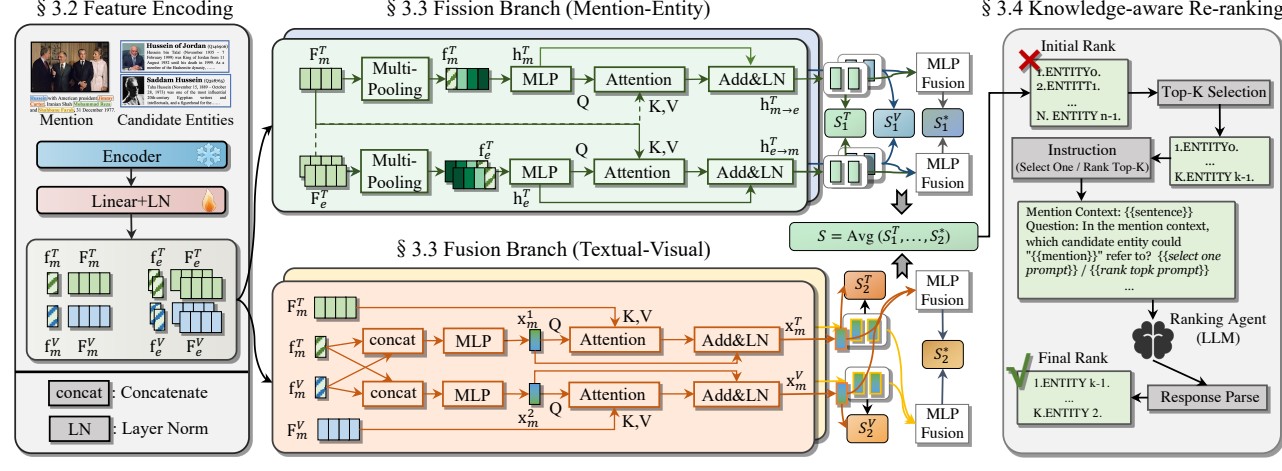

**Figure 2: Schematic illustration of the proposed FissFuse (Section 3.3) and KAR (Section 3.4).**

$\mathbf{f}_m^T, \mathbf{f}_m^V, \mathbf{f}_e^T, \mathbf{f}_e^V \in \mathbb{R}^d$, providing minimal loss of semantics while enhancing the overall contextual understanding.

## 3.3 FissFuse Framework

Previous research mainly focuses on fine-grained multimodal interactions and fusion. However, they may be affected by content discrepancy between mentions and entities. Additionally, the static encoding, where the same mention is encoded identically for different entities, exacerbates the impact of discrepancy. To address this issue, the two branches of FissFuse jointly collaborate to facilitate ample feature intertwining. In the Fission branch, our fundamental insight is to establish dynamically adaptive mention-entity representations, allowing the same mention to vary with different entities. As for the Fusion branch, we employ flexible modality complementary to alleviate content discrepancy between mentions and entities. We combine the two branches with multiple alignment scores. The overview of FissFuse framework is depicted in Fig. 2.

**3.3.1 Fission Branch.** In the Fission branch, our objective is to establish dynamic representations for each mention based on different entities, and vice versa for each entity based on different mentions. We elaborate on the overall process using textual features, and a similar procedure is applied to visual features. Specifically, we first obtain the distinctive representations of sequential features via a multi-pooling operation, as pooling offers a mapping view of original features. This process is mathematically formulated as follows:

$$\mathbf{f}_m^{T_1}, \mathbf{f}_m^{T_2} = \max(\mathbf{F}_m^T), \operatorname{avg}(\mathbf{F}_m^T),$$
$$\mathbf{f}_e^{T_1}, \mathbf{f}_e^{T_2} = \max(\mathbf{F}_e^T), \operatorname{avg}(\mathbf{F}_e^T), \tag{3}$$

where max and avg represent pooling operations along the sequence length (or image patch) dimension. Then, we integrate global features into the distribution features as follows,

$$\mathbf{h}_m^T = \operatorname{MLP}\left([\mathbf{f}_m^{T_1} || \mathbf{f}_m^{T_2} || \mathbf{f}_m^T]\right),$$
$$\mathbf{h}_e^T = \operatorname{MLP}\left([\mathbf{f}_e^{T_1} || \mathbf{f}_e^{T_2} || \mathbf{f}_e^T]\right), \tag{4}$$

in which MLP denotes multi-layer perceptron, characterized by two linear layers and ReLU activation function between them, and

$[\cdot||\cdot]$ represents concatenate operation. Thus, we derive the comprehensive contextual representation for the mention sentence and entity description. We further capture the detailed correlation and mitigate the textual content gap between mention and entities via cross-attention mechanism:

$$\mathbf{h}_{m \to e}^{T'} = \operatorname{Cross-Attention}(\mathbf{h}_m^T, \mathbf{F}_e^T),$$
$$\mathbf{h}_{m \to e}^T = \operatorname{LayerNorm}(\mathbf{h}_{m \to e}^{T'} + \mathbf{h}_m^T), \tag{5}$$

where $\mathbf{h}_{m \to e}^T$ represents textual feature of mention relative to the entity. Importantly, this suggests that the representations of a single mention vary dynamically across different entities. We also obtain the entity feature under the mention context as follows:

$$\mathbf{h}_{e \to m}^{T'} = \operatorname{Cross-Attention}(\mathbf{h}_e^T, \mathbf{F}_m^T),$$
$$\mathbf{h}_{e \to m}^T = \operatorname{LayerNorm}(\mathbf{h}_{e \to m}^{T'} + \mathbf{h}_e^T). \tag{6}$$

Similarly, following the process Eq.3 - Eq.6 by replacing textual features with visual features, we can get visual vector $\mathbf{h}_{m \to e}^V, \mathbf{h}_{e \to m}^V$. Then, we combine the two modalities features with two MLPs:

$$\mathbf{h}_{m \to e}^* = \operatorname{MLP}([\mathbf{h}_{m \to e}^T || \mathbf{h}_{m \to e}^V]),$$
$$\mathbf{h}_{e \to m}^* = \operatorname{MLP}([\mathbf{h}_{e \to m}^T || \mathbf{h}_{e \to m}^V]). \tag{7}$$

Finally, we calculate the similarity score from different perspectives:

$$S_1^T = \mathbf{h}_{m \to e}^T \odot \mathbf{h}_{e \to m}^T,$$
$$S_1^V = \mathbf{h}_{m \to e}^V \odot \mathbf{h}_{e \to m}^V, \tag{8}$$
$$S_1^* = \mathbf{h}_{m \to e}^* \odot \mathbf{h}_{e \to m}^*,$$

where $\odot$ indicates dot-product operation.

**3.3.2 Fusion Branch.** In the Fission branch, our objective is to further alleviate the content gap between mentions and entities by establishing an adaptive multimodal feature interaction, allowing semantically rich modality to complement semantically deficient one. We elaborate on the processing of mention features in detail, and a similar procedure is applied to entity features. Specifically, we first adaptively fuse global features with two different MLPs. Then, we utilize attention mechanism and skip connection to capture both

Prompt Instruction

Mention Context: {{sentence}}

Question: In the mention context, which candidate entity could "{{mention}}" refer to? Pay attention to the literal meaning, don't imagine or embellish. {{*select one prompt*}} / {{*rank topk prompt*}}

Candidate Entities (<ENTITY_ID>: <ENTITY_NAME>):

IDo: {{ENTITYo_NAME}}
ID1: {{ENTITY1_NAME}}
(more entities)

Organize response with <ENTITY_ID> and <ENTITY_NAME> in JSON format like:

{{*select one format*}} / {{*rank topk format*}}

Please directly answer the question in JSON format, and do not explain the reason.

| Select One | Rank Top-K |
|---|---|
| *select one prompt*:= Please "select" the most likely one from the following candidate entities. | *rank topk prompt*:= Please "sort" the following candidate entities according the probability from high to low. |
| *select one format*:= {"<ENTITY_ID>": "<ENTITY_NAME>"} | *rank topk format*:= {"<ENTITY_ID>": ["<ENTITY_NAME>", PROBABILITY_VALUE], ...,} |

**Figure 3: Instructions of two re-ranking strategies.**

textual-visual and visual-textual semantics, i.e.,

$$
\begin{aligned}
\mathbf{x}_m^1 &= \text{MLP}_1([\mathbf{f}_m^T || \mathbf{f}_m^V]), \\
\mathbf{x}_m^{T'} &= \text{Cross-Attention}(\mathbf{x}_m^1, \mathbf{F}_m^T), \\
\mathbf{x}_m^T &= \text{LayerNorm}(\mathbf{x}_m^{T'} + \mathbf{x}_m^1), \\
\mathbf{x}_m^2 &= \text{MLP}_2([\mathbf{f}_m^T || \mathbf{f}_m^V]), \\
\mathbf{x}_m^{V'} &= \text{Cross-Attention}(\mathbf{x}_m^2, \mathbf{F}_m^V), \\
\mathbf{x}_m^V &= \text{LayerNorm}(\mathbf{x}_m^{V'} + \mathbf{x}_m^2).
\end{aligned}
\tag{9}
$$

With the guidance of adaptively fused features, these operations imply adaptively capturing correlations between different modalities and supplementing the semantically deficient modality. Similarly, based on Eq.9 , we can also obtain relevant representations $\mathbf{x}_e^T$ and $\mathbf{x}_e^V$ for entity. Subsequently, we employ MLP fusion on both mention features and entity features, resulting in $\mathbf{x}_m^*$ and $\mathbf{x}_e^*$. After obtaining these features, we estimate the similarity between mention and entity as follows:

$$
\begin{aligned}
\mathcal{S}_2^T &= \mathbf{x}_m^T \odot \mathbf{x}_e^T, \\
\mathcal{S}_2^V &= \mathbf{x}_m^V \odot \mathbf{x}_e^V, \\
\mathcal{S}_2^* &= \mathbf{x}_m^* \odot \mathbf{x}_e^*.
\end{aligned}
\tag{10}
$$

*3.3.3 **Similarity and Loss Function.*** The symmetrical design of the two branches introduces comparable features as well as similarity measurements from different perspectives. We opt for a straightforward approach of averaging scores from these two branches as the basis for ranking candidate entities:

$$
\text{sim}_\theta(M_i, E_j) = \mathcal{S}_{ij} = \frac{1}{6} \sum_{k \in \{1,2\}} \sum_{l \in \{T,V,*\}} \mathcal{S}_k^l(i,j).
\tag{11}
$$

To minimize the model's error, we introduce cross-entropy loss to jointly optimize each similarity score, ensuring higher scores for the correct entities and lower scores for incorrect entities, which is

**Table 1: Statistics of WikiMEL and WikiDiverse.**

| Statistic | WikiMEL | WikiDiverse |
|---|---|---|
| # mention in train | 18,092 | 12,268 |
| # mention in valid | 2,585 | 1,459 |
| # mention in test | 5,169 | 1,459 |
| # mention in total | 25,846 | 15,186 |
| # candidate | 100 | 10 |
| # avg. length | 10.13 | 8.20 |

defined as follows:

$$
\begin{aligned}
\mathcal{L}(\mathcal{S}_k^l) &= -\frac{1}{|M|} \sum_{i=1}^{|M|} \log\left(\frac{\exp(\mathcal{S}_k^l(i, j^*))}{\sum_{j=1}^N \exp(\mathcal{S}_k^l(i, j))}\right), \\
\mathcal{L}_{\text{Final}} &= \sum_{k \in \{1,2\}} \sum_{l \in \{T,V,*\}} \mathcal{L}(\mathcal{S}_k^l),
\end{aligned}
\tag{12}
$$

where $\mathcal{S}_k^l(i, j^*)$ represents the score of the ground truth entity for the i-th mention. In this way, we encourage consistent judgments across different scores to facilitate the collaboration of the two branches.

## 3.4 Knowledge-aware Re-ranking

Calculating the similarity between the encoding of mentions and entities provides reliable results. However, as mentioned before, this approach is still confined to encoding rather than reasoning, and thus may not fully leverage entity knowledge. Recent advancements in LLMs have demonstrated their powerful capabilities in knowledge storage and reasoning. Inspired by this, we propose a universal and plug-and-play knowledge-aware re-ranking (KAR) method to further enhance the performance of entity linking. Specifically, given a mention, we first obtain the initial candidate entities ranking based on the scores $\mathcal{S}$ from Eq.11. Then we select top-k entities to construct instructions to ask LLM to re-rank these entities. As shown in Fig. 3, we tailor two types of instructions: one directly asking for the sorting of the top-k entities (*rank top-k*), and the other requesting the selection of the most suitable entity from the top-k entities (*select one*). Finally, we re-rank the top-k candidate entities based on the sequential order of entity IDs in the LLM's response.

## 4 EXPERIMENTS

### 4.1 Experimental Settings

*4.1.1 **Datasets.*** We conducted extensive experiments on two well-constructed publicly datasets, **WikiMEL** [34] and **WikiDiverse** [35]. WikiMEL is a human-verified dataset that is collected from Wikipedia entity pages. WikiDiverse is a manually annotated dataset collected from Wikinews, featuring a wide array of contextual topics and a diverse range of entity types. We notice some other datasets, but they are not available for some reasons. Twitter-MEL [21] used Twitter API to collect Twitter posts, but we cannot be completely reproduced due to the expiration of Twitter content. Gan et al. [9] proposed M3EL dataset containing movie reviews and movie-related images. To the best of our knowledge, they only released image features instead of raw images, which brings difficulty for comparisons with other baselines. Yang et al. [39] constructed a

Table 2: Performance comparison. * means the p-value of t-test compared with MIMIC is lower than 0.001. The best results are highlighted in bold and the second best are underlined. † denotes generative model that typically produce one result, only calculating H@1. FissFuse(B+R) means replacing the backbone with bert-base-uncased and Resnet-101.

| Dataset | WikiMEL | | | | | WikiDiverse | | | | | Avg. | |
|---|---|---|---|---|---|---|---|---|---|---|---|---|
| Metric | H@1↑ | H@2↑ | H@3↑ | MR↓ | MRR↑ | H@1↑ | H@2↑ | H@3↑ | MR↓ | MRR↑ | H@1↑ | MRR↑ |
| BERT [5] | 39.95 | 53.68 | 61.31 | 6.36 | 54.07 | 57.08 | 74.57 | 84.32 | 2.12 | 72.03 | 48.52 | 63.05 |
| BLINK [37] | 36.00 | 49.54 | 57.52 | 7.54 | 50.36 | 56.30 | 73.40 | 82.69 | 2.19 | 71.19 | 46.15 | 60.78 |
| GENRE [3]† | 60.10 | - | - | - | - | 78.00 | - | - | - | - | 69.05 | - |
| ViLT [12] | 79.40 | 84.08 | 85.65 | 3.41 | 83.80 | 40.27 | 58.17 | 68.49 | 2.91 | 58.38 | 59.84 | 71.09 |
| ALBEF [14] | 55.12 | 65.98 | 76.32 | 3.42 | 68.76 | 59.14 | 76.40 | 86.20 | 2.00 | 73.70 | 57.13 | 71.23 |
| CLIP [25] | 81.53 | 89.97 | 93.15 | 1.78 | 87.89 | 61.12 | 79.70 | 89.16 | 1.88 | 75.61 | 71.33 | 81.75 |
| DZMNED [20] | 39.41 | 50.97 | 57.90 | 7.77 | 52.13 | 29.11 | 47.37 | 61.16 | 3.53 | 49.53 | 34.26 | 50.83 |
| JMEL [1] | 47.99 | 63.60 | 71.68 | 4.33 | 62.42 | 51.55 | 68.08 | 78.49 | 2.47 | 67.15 | 49.77 | 64.79 |
| MEL-HI [42] | 30.86 | 45.26 | 54.73 | 6.22 | 47.18 | 53.88 | 70.59 | 80.00 | 2.36 | 69.01 | 42.37 | 58.10 |
| GHMFC [34] | 56.69 | 72.99 | 80.61 | 2.91 | 70.45 | 55.71 | 72.35 | 80.94 | 2.30 | 70.31 | 56.20 | 70.38 |
| DRIN [38] | 66.05 | 79.81 | 85.39 | 2.11 | 80.84 | 49.43 | 66.90 | 77.17 | 1.83 | 57.21 | 57.74 | 69.02 |
| MIMIC [18] | 81.62 | 90.29 | 93.58 | 1.77 | 88.05 | 67.90 | 85.14 | 92.63 | 1.62 | 80.57 | 74.76 | 84.31 |
| GPT-3.5† | 73.80 | - | - | - | - | 72.70 | - | - | - | - | 73.25 | - |
| GEMEL [28]† | 75.20 | - | - | - | - | 80.20 | - | - | - | - | 77.70 | - |
| GEMEL(16 shots) | 82.60 | - | - | - | - | 86.30 | - | - | - | - | 84.45 | - |
| FissFuse | 84.80* | 92.37* | 95.05* | 1.61* | 90.26* | 80.30* | 91.42* | 95.34* | 1.39* | 88.11* | 82.55 | 89.18 |
| FissFuse(B+R) | 73.68 | 85.64 | 90.48 | 2.06 | 82.78 | 72.37 | 87.51 | 92.99 | 1.57 | 83.15 | 73.02 | 82.96 |
| FissFuse+KAR | 87.89 | 93.42 | 95.36 | 1.54 | 92.02 | 83.29 | 92.53 | 95.89 | 1.35 | 89.81 | 85.59 | 90.92 |

new dataset NYTimes-MEL but the dataset is not publicly available. RichpediaMEL was built from the multimodal knowledge graph Richpedia [33], but Richpedia was no longer maintained[1], so we were unable to access the original data. We used the data processed by Xing et al. [38] instead of Luo et al. [18] because they provided data processing scripts[2] and each mention contains more candidate entities[3]. The statistics of WikiMEL and WikiDiverse are summarised in Table 1.

*4.1.2 Baselines and Evaluations.* We compared our method with three types of baselines. The text-based EL methods include BERT [5], BLINK [37] and GENRE [3]. The Vision-Language Pre-trained methods contain ViLT [12], ALBEF [14] and CLIP [25]. The MEL methods include DZMNED [20], JMEL [1], MEL-HI [42], GHMFC [34], DRIN [38] and MIMIC [18]. The LLM methods contain GPT-3.5-turbo and GEMEL [28]. Following baselines, we reported hits rate of the top-k (H@K), the mean rank (MR) among N candidate entities, and the mean reciprocal rank (MRR) among N candidate entities. The metrics are defined as follows:

$$\text{H@k} = \frac{1}{|M|} \sum_{i}^{|M|} \mathbb{I}(\text{rank}_i^N \leq k),$$

$$\text{MR} = \frac{1}{|M|} \sum_{i}^{|M|} \text{rank}_i^N, \qquad (13)$$

$$\text{MRR} = \frac{1}{|M|} \sum_{i}^{|M|} \frac{1}{\text{rank}_i^N},$$

---

[1]http://rich.wangmengsd.com/
[2]https://github.com/starreeze/drin-dataset
[3]This lead to some differences in the results compared to those reported by Luo et al. [18].

where $\mathbb{I}$ is the indicator function, $|M|$ indicates the number of mentions, $\text{rank}_i^N$ denotes the ranking for the ground truth entity of the i-th sample among N candidate entities. N is set to 100 for WikiMEL and 10 for WikiDiverse respectively.

*4.1.3 Implementations.* We initialized the encoder with pre-trained CLIP-ViT-B/32. The maximal textual input length was set to 64. The dimension $d$ of the network was set to 100. We implemented our method with PyTorch and trained the model with 2 GeForce RTX 3090 GPUs. We used AdamW [17] as the optimizer with a batch size of 128 per GPU. The number of epochs was set to 20 and 30, the learning rate was set to $4 \times 10^{-5}$ and $3 \times 10^{-5}$ for WikiMEL and WikiDiverse, respectively. We used LLaMA2-7B [31] and select one strategy for re-ranking, and K=5. We will release the code publicly after the review.

## 4.2 Performance Comparison

*4.2.1 Main Result.* Table 2 shows the numerical results. We reproduced most of the baselines, running each three times with different random seeds. The average scores are reported.

Firstly, we can see that in text-based EL methods, BERT and BLINK achieve promising results because text is fundamental modality. However, they only utilize the global sentence-level information in the text, ignoring the fine-grained semantic at the word-level. GENRE shows better results, even surpassing some MEL methods, thanks to its adoption of BART as the backbone and training with large-scale Wikipedia data. However, these methods rely solely on the text modality, failing to fully leverage rich visual semantics, which poses challenges in disambiguating in cases where textual information is limited.

Table 3: Experimental results of ablation study. The best metrics are highlighted in bold.

| Dataset | WikiMEL | | | | | WikiDiverse | | | | | Avg. | |
|---|---|---|---|---|---|---|---|---|---|---|---|---|
| Metric | H@1↑ | H@2↑ | H@3↑ | MR↓ | MRR↑ | H@1↑ | H@2↑ | H@3↑ | MR↓ | MRR↑ | H@1↑ | MRR↑ |
| FissFuse | **84.80** | **92.37** | **95.05** | **1.61** | **90.26** | **80.07** | **91.69** | **95.55** | **1.39** | **88.05** | **82.44** | **92.03** |
| (a) w/o Fission branch | 84.06 | 92.32 | 94.84 | 1.62 | 89.83 | 78.63 | 90.68 | 95.30 | 1.40 | 87.25 | 81.35 | 88.54 |
| (b) w/o Fusion branch | 83.68 | 91.78 | 94.58 | 1.68 | 89.49 | 72.67 | 88.36 | 93.49 | 1.53 | 83.58 | 78.18 | 86.54 |
| (c) w/o $\mathcal{S}_1^*, \mathcal{S}_2^*$ | 84.09 | 92.28 | 94.59 | 1.60 | 90.11 | 79.86 | 91.10 | 95.21 | 1.41 | 87.80 | 81.98 | 88.96 |
| (d) w/o $\mathcal{S}_1^T, \mathcal{S}_1^V$ | 83.83 | 91.99 | 95.15 | 1.63 | 89.72 | 75.75 | 90.27 | 94.86 | 1.45 | 85.60 | 79.79 | 87.66 |
| (e) w/o $\mathcal{S}_2^T, \mathcal{S}_2^V$ | 84.31 | 92.09 | 95.02 | 1.58 | 90.11 | 79.59 | 90.62 | 95.21 | 1.40 | 87.61 | 81.95 | 88.86 |
| (f) w/o $\mathcal{S}_1^T, \mathcal{S}_1^V, \mathcal{S}_2^T, \mathcal{S}_2^V$ | 78.97 | 88.61 | 92.46 | 1.82 | 86.30 | 75.27 | 89.11 | 94.45 | 1.49 | 85.08 | 77.12 | 85.69 |
| (g) w/o visual modality | 78.57 | 87.64 | 91.62 | 1.99 | 85.67 | 78.01 | 90.41 | 94.73 | 1.42 | 86.75 | 78.29 | 86.21 |

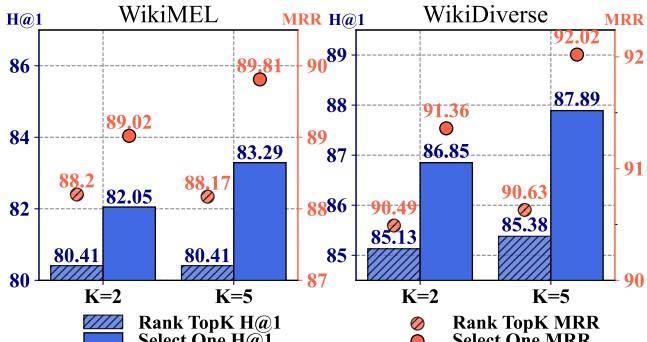

Figure 4: Results of Different strategies of re-ranking.

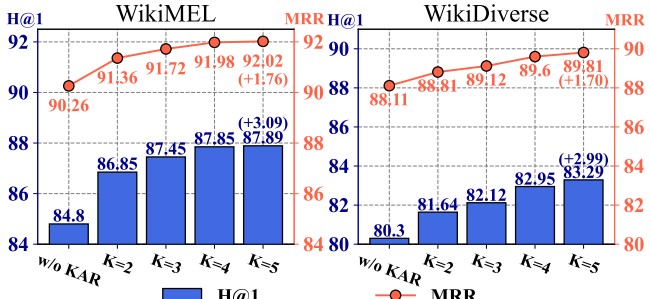

Figure 5: Results of re-ranking with different K.

Secondly, compared to text-based EL methods, VLP methods can utilize visual information, demonstrating competitive results. Specifically, benefiting from large-scale image-text pre-training, CLIP achieves the best performance in VLP methods. As competitive baseline models, we believe that these models can further improve performance by considering fine-grained semantic interactions, bridging content discrepancy, and fully leveraging multimodal information.

Thirdly, looking into MEL approaches, different methods shows a certain gap. DZMEND adopts attention to fuse features across different modalities, while JMEL uses simple concatenation and linear layers to fuse different features. Compared with GHMFC, DRIN and MIMIC, both methods show limited results. This indicates that shallow feature interaction strategies may not lead to performance improvements and could even result in degradation. In addition, although DRIN employs graph neural networks to model the interactions between mention-entities and different modalities, its performance is still lower than MIMIC. This may be due to the lack of alignment between different features, and the model does not consider fine-grained semantics, only using global features. Moreover, in terms LLM methods. GEMEL outperforms GPT-3.5 due to its fine-tuning on the MEL task and its ability to utilize visual information from images. However, this approach relies solely on the knowledge within the large language model and suffers from high complexity during inference.

Finally, the experimental results demonstrate that our proposed FissFuse achieves the second-best performance. Compared with

GEMEL, it obtains an absolute improvement of 4.85% in terms of average H@1 on WikiMEL and WikiDiverse. To eliminate the influence of the encoder, we replaced the backbone with bert-base-uncased and Resnet-101 (denoted B+R), and it still exhibits a significant advantage over GHMFC and DRIN with the same backbone. We give credit to the integration of both Fission branch and Fusion branch. Moreover, FissFuse+KAR demonstrates better performance, suggesting that the knowledge-aware re-ranking strategy can effectively further improve the MEL performance. We also conduct significant tests, and the p-values of the MRR metric on the two datasets compared with MIMIC are $3 \times 10^{-4}$ and $2 \times 10^{-6}$, respectively. Furthermore, our method exhibits better consistency on the two datasets compared to other models. These evidences validate the effectiveness of our proposed FissFuse and KAR.

*4.2.2* ***Ablation Study.*** In Table 3, we measure the impact of each component via ablation analysis. In variants (a) and (b), we remove the two branches respectively. As shown in the table, removing any one branch leads to a performance drop, highlighting the the crucial role of the well-designed mention-entity and textual-visual interaction patterns. In addition, for variants (c), (d), (e), and (f), we remove partial matching scores respectively, which leads to different degrees of decline in results on the two datasets. This indicates the need to evaluate the matching degree of mentions and candidate entities from different perspectives and semantics. We also conduct ablation on the visual modality, and the results show that introducing visual information helps to eliminate text ambiguity.

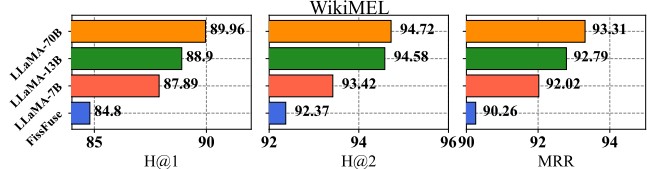

**Figure 6: Results of re-ranking with different LLMs.**

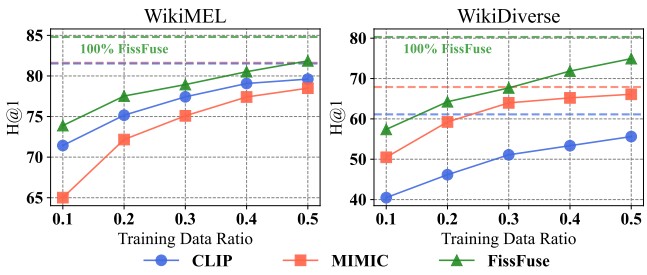

**Figure 7: Results of different scale of data. The dashed lines represent the performance the full training data.**

## 4.3 Discussion

*4.3.1* ***Re-ranking Strategy.*** In Fig. 4, We explore two re-ranking strategies, namely, *select one* and *rank top-k*. As seen, both strategies contribute to improvements, but the *select one* strategy exhibits a higher enhancement compared to *rank top-k*. Notably, even when k is set to 2 and the prompts, excluding instructions, are the same for both strategies, *select one* still performs better. This suggests that, for LLMs, executing *rank top-k* is more challenging than *select one*. This observation also aligns with human perception of the two different strategies, as ranking involves multiple rounds of comparisons, assuming equal importance for each candidate entity, while selection only requires identifying the most probable answer.

*4.3.2* ***Impact of Top-K and LLM***. In Fig. 5 and Fig. 6, we further examine the impact of Top-K selection during re-ranking and the influence of the size of LLMs. With an increase in K, we observe a gradual improvement in performance, indicating that more candidate entities contribute to the LLMs making correct selections. Additionally, as K increases, the degree of performance improvement gradually diminishes. We suspect this might be approaching the performance boundary of LLM, influenced by the model's scale. As seen in Fig. 6, with an increase in model size, the re-ranking performance further improves, thus confirming that the capability strengthens as the model size increases.

*4.3.3* ***Scale of Data***. Considering that collecting high-quality labeled data is expensive and time-consuming, we also investigate the impact of data scale on performance, and the results are shown in Fig 7. The dotted line represents the model performance using 100% of the training data, and the data points represent the performance of models using different proportions of training data.It can be observed that using only 10% of the training data results in significant performance degradation for both CLIP and MIMIC, indicating the necessity of sufficient data. With an increase in training data, all

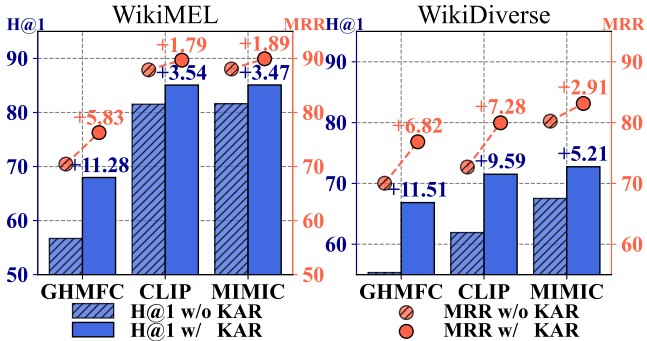

**Figure 8: Results of re-ranking with different MEL models.**

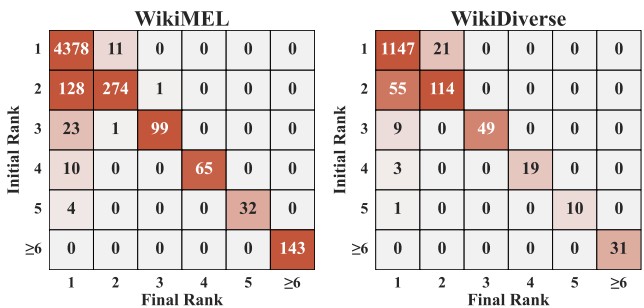

**Figure 9: Error Analysis of Re-ranking. We use LLaMA2-7B and Select One strategy, K=5. The color shades reflect the magnitude of values.**

models consistently show improvement. FissFuse consistently maintains the best performance, demonstrating its adaptability even in low-resource scenarios.

*4.3.4* ***Generality of Re-ranking Strategy***. We take a further step, investigating the generality of re-ranking across different MEL methods. As shown in Fig. 8, we select three baselines (i.e., GHMFC, CLIP, MIMIC). The results clearly demonstrate significant improvements for each baseline, with notably larger enhancements observed for weaker models (GHMFC) compared to the strong model (MIMIC). Furthermore, it can be seen that despite the performance boost from re-ranking, the relative order among these methods remains unchanged (MIMIC > CLIP > GHMFC). This indicates that, for the MEL task, re-ranking by LLM is not the sole determinant, and a robust initial MEL method is equally crucial for overall performance. Different from GEMEL requiring fine-tuning, the observation confirms the seamless generality and effectiveness of KAR with different MEL methods.

*4.3.5* ***Error Analysis of KAR***. We conduct error analysis for KAR on two datasets. Fig 9 shows the matrix of ranking changes for ground truth entities before and after re-ranking. Specifically, the element in the first row and first column represents the number of samples where the ground truth entity is ranked first initially among all candidate entities and remains ranked first after re-ranking. The elements on the diagonal represent the samples whose ranking remains unchanged, the elements above the diagonal represent

| Mention | Ground Truth | FissFuse | MIMIC | CLIP | GHMFC |
|---|---|---|---|---|---|
| **Wouter Van Bellingen** with on his left side mayor Freddy Willockx. | Wouter Van Bellingen — Wouter Van Bellingen (born 20 April 1972) is a Rwandan born black politician…… | 1. Wouter Van Bellingen 2. Janwillem van de Wetering 3. Martijn van Helvert | 1. Martijn van Helvert 2. Wouter Van Bellingen 3. Nobel Memorial Prize in Economic Sciences | 1. Nobel Memorial Prize in Economic Sciences 2. Martijn van Helvert 3. Wouter Van Bellingen | 1. Wouter van der Goes 2. Wouter Van Bellingen 3. Martijn van Helvert |
| STS-126 crew (left to right): Magnus, Bowen, Pettit, **Ferguson**, Boe, Kimbrough and Stefanyshyn-Piper. | Christopher Ferguson — Christopher J. "Fergy" Ferguson (born September 1, 1961) is a Boeing commercial astronaut and …… | 1. Christopher Ferguson 2. Ferguson, Missouri 3. Ferguson, Iowa | 1. Crew 2. Christopher Ferguson 3. Ferguson, Iowa | 1. Crew 2. Christopher Ferguson 3. Ferguson, Iowa | 1. Ferguson, Missouri 2. Christopher Ferguson 3. Ferguson-Brown Company |

Figure 10: Qualitative results are shown with one instance per row. We display top 3 ranked entities for each method. The highlighted text in the Mention context represents the mention words. Each entity contains an image, a name, and a description. To better showcase the result, we omit the descriptions of the top 3 ranked entities.

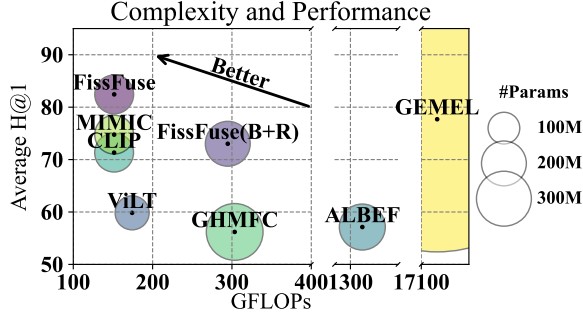

Figure 11: Complexity and performance comparison. Larger circles indicate more parameters. The y-axis shows the average H@1 on two datasets, and the x-axis represents the FLOPs required to infer one sample.

the samples whose ranking is degraded, and the elements below the diagonal represent the samples whose ranking is improved. It can be seen that the ranking of most samples remains unchanged. Besides, the number of samples with improved ranking is much larger than the number of samples with degraded ranking, which verifies the effectiveness of KAR. For the samples with degraded ranking, one possible reason is the inductive bias of large models when the text information is limited. We believe that adding visual information may help to correct this bias, which is also a direction we will explore in the future.

*4.3.6* ***Complexity Analysis***. In Fig. 11, we further compare the efficiency of various MEL methods. We calculate the number of floating-point operations (FLOPs) required during inference and the total number of the model's parameters. We set the number of candidate entities to 10 and calculate the FLOPs needed when inference. It can be seen that our FissFuse achieves significantly better results compared to MIMIC and CLIP, despite having similar amounts of parameters and FLOPs. GEMEL heavily relies on the

capabilities of LLMs, inevitably leading to high complexity and computational overhead. This observation demonstrates that our framework has advantages in both efficiency and performance.

*4.3.7* ***Qualitative Result***. Finally, to gain a more intuitive understanding of the advantages of our model, we empirically analyze real cases in Fig. 10. As evident from both cases, the issue of content discrepancy arises when a mention contains multiple objects in either sense or text, while the entity is focused on a single object. Specifically, in the first case, CLIP incorrectly matches the image of *Nobel Memorial Prize in Economic Sciences* with the mention image, despite their apparent similarities. In the second case, both MIMIC and CLIP tend to overly focus on visual information, mistakenly associating *Crew* with *Ferguson*. However, the mention image conveys the theme of *spaceflight* and *astronauts*, instead of direct visual indications. This phenomenon indicates that these models do not effectively handle the relationship between mention-entity and textual-visual features. In contrast, our proposed FissFuse considers the dynamic mention-entity and cross-modal interactions, thereby alleviating the issue of content discrepancy.

## 5 CONCLUSION

In this paper, we proposed a novel framework FissFuse as well as a knowledge-aware re-ranking method KAR to fill the gap in content and knowledge for multimodal entity linking. Specifically, FissFuse collaborates two branches to establish dynamic, adaptive feature interactions, alleviating content discrepancy between mentions and entities. Additionally, KAR leverages the entity knowledge and reasoning capabilities of LLMs for re-ranking. Extensive experiments on two public datasets have validated the effectiveness of FissFuse and KAR.

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
