# OpenReview forum: "Bridging Gaps in Content and Knowledge for Multimodal Entity Linking"
_acmmm.org/ACMMM/2024/Conference — MM2024 Poster_

### Official Review · Reviewer_KNbo · 2024-05-11

**Rating:** 4
**Confidence:** 4

**Summary:**

This paper focuses on the task of Multimodal Entity Linking and introduces targeted methodological designs. The proposed approach includes feature modeling, fusion components (FissFuse), and an LLM-based re-ranking section (KAR).

The paper presents a wealth of detailed experiments, including main experiments, ablation studies, and case studies, which demonstrate the superiority of the methods.

**Strengths:**

1. The paper focuses on the Multimodal Entity Linking (MEL) task, a common and important task with rich application value.

2. The architectural design of the methods introduced in the paper is novel, especially the use of Large Language Models (LLMs) for re-ranking, which is a reasonable approach.

3. The selection of baselines is appropriate and the experiments conducted are extensive.

**Limitations:**

1. The discussion on Content Discrepancy and Insufficient Knowledge Utilization challenges is not entirely convincing. Most existing methods in MEL focus on addressing these very challenges by extracting valuable information from content that may have discrepancies, contrary to what is stated in the paper that these issues are overlooked. I recommend that the authors refine their discussion on these challenges and the motivation of the study to make the research value clearer to the readers.

2. Regarding efficiency, the use of LLMs for re-ranking might impact the time and token consumption during MEL. Although Section 4.3.6 discusses the complexity of the methods, I suggest adding a comparative analysis of time consumption for different models, as well as statistics on token consumption. This is crucial for a comprehensive evaluation of the method’s value.

3. The utilization of LLMs seems limited (LLAMA). Could the authors supplement the experiments with a discussion on the effects of using multimodal LLMs or other LLMs for MEL?

**Suitability:**

2

---

### Official Review · Reviewer_4Wkw · 2024-05-20

**Rating:** 4
**Confidence:** 3

**Summary:**

This paper identifies two primary issues in the MEL task:(1) Content Discrepancy, which refers to the significant differences between the multimodal contexts of mentionsand entities, and (2) Insufficient Knowledge Utilization, where most MEL methods tend to extract knowledge from PLMs but are limited to using PLMs for encoding rather thanreasoning. To address these issues, the authors propose theFissFuse framework:
1. FissFuse dynamically integrates the contexts ofmentions and entities using a cross-attention mechanism.
2. The KAR module within the FissFuse framework leverages the powerful reasoning capabilities of LLMs through instruction templates to re-rank the retrieved entities.

**Strengths:**

1. This paper is fully experimented and discussed.
2. This paper is well written.
3. The two questions posed in this paper are insightful.

**Limitations:**

1. Fission branch and fusion branch are very complex, i'm curious why the authors designed them this way instead of using transformer blockor simple cross attention?
2. LLMs don't handle id type information particularly well, can the author discuss this aspect in more detail?

**Suitability:**

3

---

### Official Review · Reviewer_aV3Y · 2024-05-23

**Rating:** 2
**Confidence:** 4

**Summary:**

This paper proposes a Multimodal Entity Linking (MEL) method called FissFuse and a knowledge-aware re-ranking strategy called KAR. FissFuse uses features of different modalities, sources, and granularities through two branches (Fission Branch and Fusion Branch) to alleviate the content discrepancy. Extensive experiments demonstrate that the proposed method performance on public datasets.

**Strengths:**

1. The paper proposes a dynamic and adaptive feature interaction mechanism that alleviates the content discrepancy between mentions and entities.
2.Comprehensive evaluations and ablation studies validate the effectiveness and generality of the proposed FissFuse and KAR
3. The experiments are richer and provide more substantial insights.

**Limitations:**

1.The motivation of the paper is not clearly articulated. There are no obvious examples or supporting evidence provided to justify the focus on content discrepancy.
2.In the introduction, although this paper claims that pictures can assist in character classification, I do not clearly see the necessity of pictures in figure 1.
3. The novelty of the article is limited. I do not find any significant difference between the proposed Fission Branch and the Fusion Branch, and the main model is cross-attention, which is well-known and used in various multi-modal tasks.
4. Some standard academic writing conventions should be followed. For instance, necessary explanations should be provided when terms are mentioned for the first time, such as KAR.
5. I noticed that a significant improvement is attributed to KAR, and I am curious whether a multi-modal LLM with a few-shot setting could achieve similar results. I would like to see an exploration of this possibility.
6.In Table 2, FissFuse (B+R) shows lower performance compared to the results without the visual modality in Table 3, which seems unreasonable to me. Without the visual modality, there should be no FissFuse module. I hope my understanding is incorrect.

**Suitability:**

3

---

### Official Review · Reviewer_4NxW · 2024-05-26

**Rating:** 4
**Confidence:** 3

**Summary:**

The paper introduces the FissFuse framework, which dynamically enriches multimodal feature representations through Fission and Fusion branches. It also develops the KAR method, utilizing LLMs for knowledge reasoning to improve the reranking process in entity linking tasks. Experiments on two datasets demonstrate the effectiveness of FissFuse and KAR in MEL tasks.

**Strengths:**

The FissFuse framework dynamically adapts to different perspectives through its two branches, Fission and Fusion, which is a novel approach in the MEL task. The KAR method leverages the internal knowledge and reasoning capabilities of large language models (LLMs), potentially enhancing the accuracy of entity linking.

Experimental Validation: The paper conducts extensive experiments on two publicly available MEL datasets, demonstrating the effectiveness of FissFuse and KAR.

Compared to existing baseline methods, FissFuse achieves significant performance improvements across multiple evaluation metrics.
KAR, as a plug-and-play module, seamlessly integrates with existing LLMs, enhancing the performance of MEL tasks and showcasing good generalizability.

The paper presents thorough experiments and detailed analyses, with well-prepared figures and tables.

**Limitations:**

The KAR method is highly dependent on the performance of LLMs, and only the performance of llama2-7B was tested.

The paper does not provide a detailed discussion on the computational resource requirements of the FissFuse and KAR methods, which might be an issue in resource-constrained environments.

Before completing entity linking (EL), the mention is uncertain about which entity it corresponds to. Why is a dynamic feature necessary for the mention-entity pair? If it is for all candidate entities, does the effectiveness of the dynamic feature heavily depend on the preparation of candidate entities?

There are doubts about the performance. On WikiMEL, the officially reported Top-1 performance of GHMFC is 43.6, so why is H1 reported as 76 here? Is there any difference between H1 and Top-1 in metrics? I have doubts about the reported performance, and if my concerns are addressed, I might consider raising the score.

Several other MEL methods have not been considered, such as "Visual Entity Linking via Multi-modal Learning" and "A Dual-way Enhanced Framework from Text Matching Point of View for Multimodal Entity Linking".

The significance of the main model diagram is difficult to understand. It would be better to redraw it, as it even made me consider rejecting the paper. The diagram lacks a legend.

The innovation of KAR is insufficient. My understanding is that your innovation is essentially an engineering application of the capabilities of large models, using them for reranking and filtering based on text comprehension.

**Suitability:**

3

---

### Meta-Review · Area_Chair_Pbfq · 2024-06-28

**Recommendation:** Accept (Poster)
**Confidence:** 5

**Metareview:**

The paper introduces the FissFuse framework, which dynamically enriches multimodal feature representations through Fission and Fusion branches, and develops the KAR method, leveraging LLMs for knowledge reasoning to improve entity linking tasks. This approach is applied to Multimodal Entity Linking tasks and validated on two datasets, demonstrating significant performance improvements over existing methods. The reviewers appreciated the innovative aspects of FissFuse and KAR, highlighting the dynamic adaptation of feature interactions and the effective use of LLMs for re-ranking. The comprehensive experimental validation and detailed analyses further strengthened the paper's contributions. However, some concerns were raised about the dependency on LLM performance, the lack of discussion on computational resources, and the need for clearer motivation and explanation of certain components. Despite these concerns, the authors provided satisfactory responses during the rebuttal phase, addressing the main issues. Given the novel contributions, comprehensive validation, and the overall positive assessment from the reviewers, the recommendation is to accept the paper.